# Integrative Analysis of Fecal Metagenomics and Metabolomics in Colorectal Cancer

**DOI:** 10.3390/cancers12051142

**Published:** 2020-05-02

**Authors:** Marc Clos-Garcia, Koldo Garcia, Cristina Alonso, Marta Iruarrizaga-Lejarreta, Mauro D’Amato, Anais Crespo, Agueda Iglesias, Joaquín Cubiella, Luis Bujanda, Juan Manuel Falcón-Pérez

**Affiliations:** 1Exosomes Laboratory, CIC bioGUNE, 48160 Derio, Spain; marc.clos.garcia@sund.ku.dk; 2Biodonostia, Grupo de Enfermedades Gastrointestinales, 20014 San Sebastian, Spain; luis.bujandafernandezdepierola@osakidetza.eus; 3Biodonostia, Grupo de Genética Gastrointestinal, 20014 San Sebastian, Spain; koldo.garcia@biodonostia.org (K.G.); mauro.damato.mda@gmail.com (M.D.); 4Centro de Investigación Biomédica en Red de Enfermedades Hepáticas y Digestivas (CIBERehd), 08036 Barcelona, Spain; joaquin.cubiella.fernandez@sergas.es; 5OWL Metabolomics, Bizkaia Technology Park, Derio, 48160 Bizkaia, Spain; calonso@owlmetabolomics.com (C.A.); miruarrizaga@owlmetabolomics.com (M.I.-L.); 6IKERBASQUE, Basque Foundation for Sciences, 48013 Bilbao, Spain; 7School of Biological Sciences, Monash University, Clayton VIC 3800, Australia; 8Department of Gastroenterology, Complexo Hospitalario Universitario de Ourense, Instituto de Investigación Sanitario Galicia Sur, 32005 Ourense, Spain; anais.crespo.lois@sergas.es (A.C.); agueda.iglesias.gomez@sergas.es (A.I.); 9Metabolomics Platform, CIC bioGUNE, 48160 Derio, Spain

**Keywords:** metabolomics, microbiome, omics, multiomics, omics integration, integration

## Abstract

Although colorectal cancer (CRC) is the second leading cause of death in developed countries, current diagnostic tests for early disease stages are suboptimal. We have performed a combination of UHPLC-MS metabolomics and 16S microbiome analyses on 224 feces samples in order to identify early biomarkers for both advanced adenomas (AD) and CRC. We report differences in fecal levels of cholesteryl esters and sphingolipids in CRC. We identified *Fusobacterium, Parvimonas* and *Staphylococcus* to be increased in CRC patients and Lachnospiraceae family to be reduced. We finally described *Adlercreutzia* to be more abundant in AD patients’ feces. Integration of metabolomics and microbiome data revealed tight interactions between bacteria and host and performed better than FOB test for CRC diagnosis. This study identifies potential early biomarkers that outperform current diagnostic tools and frame them into the stablished gut microbiota role in CRC pathogenesis.

## 1. Introduction

Colorectal cancer (CRC) represents almost 10% of global cancer incidence [1], being the second leading cause of death in developed countries [2]. CRC can develop sporadically or in the context of inflammatory processes [3] and has been shown to be highly influenced by lifestyle factors, like diet [4] and physical activity [5]. Nowadays, non-invasive massive screening for CRC is done through fecal occult blood (FOB) test, which is the gold standard for the diagnosis, even though it has been shown to have high sensitivity but low specificity [6]. One of the current challenges of CRC treatment is its high degree of heterogeneity. It is unknown why CRCs of a similar stage, being histologically indistinguishable behave differentially, both in recurrence and chemotherapy response [1]. Therefore, there is a need for new molecular parameters that are able to distinguish between CRC types for a better treatment outcome [1]. In this context, metabolomics of feces samples have provided new non-invasive, accurate and predictive biomarkers for CRC, as our group and other authors have suggested [7,8,9]. Metabolomics is the omics technology dedicated to the study of the metabolome, the complete set of low molecular weight (<2000 Da, a.k.a metabolites) that are context-dependent [10]. Relevant technological advances and new analytical and bioinformatics tools development allow the measurement of large metabolite numbers simultaneously. Metabolomics, thus, has become a relevant technology for biomarker identification in a range of diseases, including cancer [11,12,13,14]. Importantly, the reported CRC-metabolomics studies have identified a list of metabolites systematically altered in CRC, that are involved in carbohydrate, amino acids and lipid-related metabolic pathways, including tricarboxylic acid (TCA) cycle and short-chain fatty acids (SCFA) that could be responsible for promoting tumoral growth and progression. Among the factors that influence these metabolisms, and may be related to CRC development and progression, the microbiota has been recurrently pointed to be one of them and suggested the bacterial driver-passenger hypothesis for CRC initiation and progression [15,16]. This hypothesis is highly supported by the fact that germ-free mice models susceptible to CRC have fewer tumors that non-germ-free mice [17,18,19]. This hypothesis arises from the adenoma-carcinoma sequence model developed before that states that the accumulation of mutations and epigenetic alterations promote epithelial hyperplasia in the colon (adenoma) that later results in CRC [20,21]. Mutations have been reported to occur in tumor suppressor genes (APC, CTNNB1, DCC, P53) and oncogenes (KRAS, MYC), proposing to start in APC gene, leading to adenoma and finishing with P53 causing the transition to CRC. Driver-passenger bacteria hypothesis suggests that the first epithelial transformations may be caused by certain intestinal bacteria. This epithelial damage leads to a change in the pre-tumoral microenvironment that favors the colonization of the region by opportunistic bacteria, the passenger ones, that outgrowth and replaces the driver bacteria [16]. Among proposed driver bacteria there are Bacteroides, Shigella, Citrobacter, Salmonella and *E. coli*, while for passenger bacteria the following have been reported *Fusobacterium* spp., *Streptococcus gallolyticus*, *Clostridium septicum*, Coriobacteriaceae, Roseburia and Faecalibacterium genera. Full understanding on how the bacteria composition impact in host metabolism is still lacking. Therefore, further investigation connecting microbiome and metabolome are needed.

In the current work, we have combined data from targeted UPLC-MS metabolomics and V1–V2 16S rDNA sequencing from the same stool samples for 77 healthy controls, 69 advanced adenomas and 99 CRC patients in order to identify potential non-invasive, early biomarkers for disease progression.

## 2. Results

### 2.1. Clinical Samples

Three clinical sample batches were used in this study, coming from COLONPREDICT study [7,22] and the Biobank of the Instituto de Investigación Sanitaria Galicia Sur. This study was a multi-center, cross-sectional blinded study designed to generate new CRC diagnostic tests in symptomatic patients based on available biomarkers, clinical and demographical data, approved by the Clinical Research Ethics Committee of Galicia (Code 2011/038). The distribution of samples in each data collection batch and clinical status is summarized in Figure 1.

Male individuals presented with major prevalence of both AD and CRC when compared to healthy group (62.23%, 59.60% vs. 44.74%) and they were older that control group (67.99, 70.16 vs. 64.62 years old respectively). Both group of patients also presented with higher presence of FOB (median 133 ng/mL in AD, 681 cng/mL in CRC) compared to controls (median 15 ng/mL), as well as CEA (median 1.45 ng/mL (C), 1.6 ng/mL (AD), 4.25 ng/mL (CRC) and COLONPREDICT risk score (median 0.01 (C), 0.05 (AD) and 0.5 (CRC)).

From this cohort, we obtained a total of 245 samples for the metabolomics and 224 samples for the microbiome analysis (Appendix A). We first performed UPLC-MS analysis of the fecal samples, applying both multivariate and univariate analyses. Afterwards, we analyzed the microbiome of the same samples and later we combined both datasets to characterize the alterations identified by every single omics and to provide with a potential diagnostics model that combined both data types.

### 2.2. Metabolomics Analysis

Considering all samples together, we performed several comparisons: Control vs. Case samples (both AD and CRC combined), C vs. AD, C vs. CRC and AD vs. CRC sample groups. Principal Component Analysis (PCA) (Figure 2A) did not identify any specific clustering of samples. Partial Least Squares discriminant analysis (PLS-DA), though, was able to discriminate the CRC samples from the other two groups, which did not present significant differences between themselves (Figure 2B). This discrimination capability was confirmed by the analysis of the models accuracy, which revealed that only the C vs. CRC PLS-DA model was able to significantly discriminate between those groups (ANOVA *p*-value 0.013), while the other models did not demonstrate that capacity (ANOVA *p*-value for C vs. AD 1 and AD vs. CRC 0.2). The three groups PLS-DA model did not show any discrimination capability neither (*p*-value = 1). PLS-DA loadings plot showed that the most contributing metabolites to this separation were mainly cholesteryl esters (ChoE) and sphingomyelins, with a certain influence of glycerophosphatidylcholine (PC) species. PLS-DA analysis in a pairwise fashion showed that CRC was clearly differentiated when compared both to C and AD groups (Figure 2C). The comparison between C and AD showed a less-clear separation between C and AD (Figure 2C).

In respect to the univariate analysis, we found that CRC samples presented mostly the same differential metabolites when compared to either C or AD sample groups (Figure 2D). C and AD group comparison, instead, revealed that these two groups did not present significant metabolic differences, although a certain trend on AD presenting higher levels of triacylglycerol metabolite species was identified.

Full identifications of the differential metabolites per group can be found in Appendix A. Thus, CRC samples presented generalized higher levels for both ChoE and sphingomyelins classes metabolites than C and AD samples. Between AD and CRC we also identified generalized higher levels of PCs metabolites in the latter samples and also differences in diacylglycerol metabolic species. Relevantly, these results were in agreement with our CRC fecal metabolomics study [7], which showed similar alterations.

We then analyzed how the metabolomics results could be associated with clinical metadata, identifying triacylglycerol species to negatively correlate with age and ChoE and sphingomyelins to correlate with FOB and calprotein measurements. Mapping of metabolites to different databases with IMPaLA revealed a significant amount of altered metabolic pathways between CRC and the other sample groups (Appendix A). Among them, several pathways related with lipid metabolism, and also, pathways related to immune system activation and to pathogenic *Escherichia coli* infection, which may be associated with microbiota alterations in CRC patients. Mapping of identified metabolites to *E. coli* KEGG database identified them to be related mostly to 3 pathways, two of them related to lipid (sphingolipids and glycerophospholipids) metabolisms and the cationic antimicrobial peptide (CAMP) resistance pathway, thus suggesting a potential association with bacterial membrane components.

### 2.3. Microbiome Analysis

For this analysis, DNA was obtained from 231 stool samples (77 controls (C), 65 AD and 89 CRC). Although a different average of lyophilized stool sample was used per sample group ${\bar{X}}_{C}=$ 67.9 mg, ${\bar{X}}_{\mathrm{AD}}=$ 70.27 mg, ${\bar{X}}_{\mathrm{CRC}}=$ 78.58 mg), no difference was observed for this factor (ANOVA *p*-value 0.087). Seemingly, no differences were observed neither for the DNA concentration obtained and the sample group (ANOVA *p*-value 0.657, $\overset{´}{\chi }$C = ${\bar{X}}_{C}=$ 264.35 ng/µL, ${\bar{X}}_{\mathrm{AD}}=$ 253.09 ng/µL, ${\bar{X}}_{\mathrm{CRC}}=$ 277.66 ng/µL). No correlation was observed between the initial amount of sample used and the DNA concentration obtained (*rho*-0.038, *p*-value 0.566) (Appendix A). 224 samples were correctly sequenced, generating a total of 7,762,116 reads distributed in 34,652.30 sequences/sample in average (minimum = 11,371, maximum = 73,019, median = 33,574). After demultiplexing and quality control steps, 6,221,946 sequences remained in the study (80.37%). These sequences were distributed in 17,641 features (operational taxonomic units, OTUs) (Appendix A). Sequencing data was uploaded to ENA repository (PRJEB33634) and OTUs table can be found in Appendix A.

We analyzed both α and β diversities by using distinct indexes. While PCoA performed upon Bray-Curtis distance index measured in unannotated OTUs composition did not show any specific clustering of samples by diagnostics (Figure 3A), PERMANOVA analysis upon Bray-Curtis distance matrix showed that the stool microbiome composition between sample groups was indeed different (*p*-value 0.001, pseudo-F = 1.303). Pairwise PERMANOVA showed that, specifically, stool microbiome was able to differentiate CRC sample group from the other two (*p*-value, *q*-value and pseudo-F: C vs. CRC 0.001, 0.001, 1.502; AD vs. CRC 0.001, 0.001, 1.413), while no difference was observed between C and AD stool microbiome compositions (*p*-value 0.705, *q*-value 0.705, pseudo-F 0.951). Bray-Curtis analysis performed upon genera-annotated microbiome composition showed the same results. Notably, supervised PLS-DA analysis was able to completely discriminate between each sample group included, consistent with the PCoA PERMANOVA results (Figure 3B). The loadings of the PLS-DA revealed a broad distribution of both Firmicutes and Bacteroidetes OTUs, spanning towards the three sample groups, as expected (Appendix A). Removing these OTUs (2571/2953 total OTUs) from the loadings plot revealed Fusobacteria to be mainly contributing to CRC samples differential clustering observed in the Figure 3B (Appendix A).

In agreement, alpha diversity measurements revealed the same pattern as described above. While CRC presented differences when compared to the other two group samples, C and AD did not present differences in microbiome composition richness. Interestingly, CRC microbiome composition was found to be richer, with higher different OTUs identified than the other C and AD sample groups. Either way, a more equilibrated diversity index as Shannon one was found to be non-significant for any of the sample groups (Figure 3C), thus suggesting that although more different genera were identified for the CRC samples group, no bacteria prevailed above others.

Afterwards, we performed taxonomical analysis of the 17,641 OTUs, and only 645 were unclassified, at least at the phylum level (3.66%). No Archaea bacteria were identified in our analysis. Thus, among the classified OTUs, we identified 15 phyla, 27 classes, 45 orders, 77 families, 172 genera and 166 species. We decided to study the differential abundances of different phyla and genera between the three sample groups. First, we studied which phyla were differentially abundant between the three sample groups included in the study by ANOVA test, identifying three phyla that met these criteria: Bacteroidetes, Firmicutes and Fusobacteria. To better clarify the origin of these abundance differences, we used Tukey’s HSD test that identified that most of the differences were due to CRC sample group (Table 1). Thus, all three phyla were differentially abundant in CRC when compared to C, while only two (Bacteroidetes and Firmicutes) presented statistically significant differences between C and AD groups. Fusobacteria did not present differences between C and AD groups. No phyla were identified to be different between AD and CRC microbiome compositions. 

All three sample groups exhibited a similar pattern of phyla abundance. Thus, the majority of the microbiome population was identified to be from Firmicutes phylum, although this was reduced in both AD and CRC patients. Bacteroidetes was the second most abundant phylum, being increased in AD and CRC patients when compared to controls. After that, Proteobacteria and the other phyla followed (Figure 3D and Appendix A). Interestingly, Fusobacteria phylum was mainly found in CRC population, with a nearly null abundance in both C and AD sample groups (Figure 3E). Finally, we studied the changes in the Firmicutes:Bacteroidetes ratio, which was found to diminish in AD and CRC patients (Figure 3E) and reported to be altered in metabolic diseases [23].

To identify differences at genus level, we employed SIAMCAT tool to test if we could identify any association between bacteria genera and the following confounding factors: gender, sample batch and FOB. Using an adjusted 0.05 significance threshold, we could not identify any genus associated with the sample gender. When analyzing the potential associations to FOB measurements, we found that two genera could be associated with FOB concentration, *Parvimonas* and *Peptostreptococcus* (Appendix A). For the differences between sample batches, we found that for the sample batch 3 several genera were significantly differentially abundant: *Staphylococcus, Bifidobacterium, Clostridiaceae 02d06, Megasphaera, Peptostreptoccaceae Clostridium, Odoribacter* and *Synergistes,* although the most evident one was found for the *Staphylococcus* (Appendix A). We then tried to identify potential differences in the abundance of genera between clinical sample groups. To this aim, we performed three comparisons: C vs. CRC, C vs. AD and AD vs. CRC (Appendix A). No significant differences in the genera abundance of stool microbiome were found between healthy controls and adenoma groups.

Finally, we also studied the differences between the three sample groups at genus level by means of compositional data analysis, using ALDEx2 R package [24]. C vs. AD patients did not present any difference at the significance level used (adj. *p*-value 0.05) (Figure 4A). In agreement with the previous approach using SIAMCAT tool, C vs. CRC comparisons showed that three genera were overrepresented in CRC patients, at adjusted *p*-value 0.05 significance level, *Fusobacterium, Staphylococcus* and *Parvimonas,* while 4 genera were found to be reduced in those same patients, three *Lachnospiraceae* genera (*Coprococcus, Blautia* and *Clostridium*) and *Streptococcus* (Figure 4B). 

For the AD vs. CRC comparison, an increased abundance of both *Staphylococcus* and *Parvimonas* genera was identified, while *Fusobacterium* was not significantly different, although there was still a trend to be higher in CRC (*t*-test adjusted *p*-value 0.064, Wilcoxon adjusted *p*-value 0.059). Reduced in CRC, we found again three *Lachnospiraceae* genera, the same *Coprococcus* and *Blautia* genera as before and *Dorea* which was not found to be significantly between C and CRC comparison. We also found *Coriobacteriaceae Adlercreutzia* genus to be underrepresented in CRC patients when compared to AD individuals, which had not been identified as different in the comparison between C and CRC patients (Figure 4C). A summary of the identified different abundances by ALDEx2 technique is presented in Figure 4D.

In summary, combining the data of both ALDEX2 and SIAMCAT approaches, 16 genera were found to be differential for some of the three sample groups. We studied how the abundances of these 16 bacterial genera changed depending on the disease progression, from healthy controls to the last CRC stage. We divided the CRC samples into five groups, depending on the CRC stage at sample collection moment. Based on the clinical data available (Appendix A), the number of samples per group to perform the comparative analysis was: 74 C, 62 AD, 3 CRC-0, 22 CRC-I, 22 CRC-II, 30 CRC-III and 6 CRC-IV. This analysis confirmed that AD sample group had a higher relative abundance of *Adlercreutzia* genus when compared to both C and CRC sample groups (Figure 5). For the majority of genera elevated in CRC patients (*Bulleidia, Fusobacterium, Butyrivibrio, Peptostreptococcus, Staphylococcus, Parvimonas* and *Selenomonas*) we identified a trend in which all these genera increased with the worsening of the disease, thus in final stages they were more abundant than in earlier CRC stages. An inverse trend was found for the genera decreased in CRC patients, mostly from the *Lachnospiraceae* family, for which the relative abundance decreased from healthy controls to each step of the disease, including AD group. Finally, *Streptococcus* was reduced in CRC when compared to both C and AD samples (Figure 5).

Finally, we used PICRUSt2 tool to identify potential differences in the metabolic capability of each sample group’s microbiome. PCA analysis on the pathway abundances did not show any difference between sample groups, although a tendency for the CRC samples group to cluster differently was observed (Appendix A). Multivariate and univariate identified several pathways contributing to this separate clustering, including amino acids biosynthesis; fermentation (to isobutanol, acetate and lactate); glucose-related pathways (gluconeogenesis, gycolysis and glycogen biosynthesis and degradation); saturated fatty acids elongation; reductive incomplete TCA; nitrate-degradation and methanogenesis-related pathways (Appendix A).

### 2.4. Combination of Microbiome and Metabolomics Data

#### 2.4.1. MixOmics

We used the CLR normalized genus data and the log-normalized metabolomics data as input for mixOmics pipeline. Using this tool, we generated a sparse block PLS-DA analysis combining both omics datasets in order to analyze the discriminative capability of combined data (Figure 6A) and both of them separately (Figure 6B). 

In both cases, we saw that all sample groups presented a highly diverse population, although CRC samples tended to separate from the other two sample groups and cluster together. The sPLS-DA performed combining both omics datasets shows that C and AD samples occupied the same space, thus reflecting fewer differences between those samples, while CRC samples occupied a more diverse range of space on the positive region of the first component (Figure 6A). Individual PLS-DA showed that sample groups distribution was relatively different depending upon the dataset analyzed. Thus, while microbiome data was unable to discriminate between C and AD samples at all, overlapping both groups, metabolomics showed a reduced ability to differentiate between C and AD sample groups. Both technologies were able to discriminate quite well the CRC samples from both C and AD sample groups (Figure 6B).

Then, we decided to study the interconnections between the metabolomics and metagenomics data by using HAIIA, a tool dedicated to identify both linear and non-linear associations between two distinct datasets [25]. In agreement with the mixOmics analysis, HAllA identified several genera that correlated with different metabolites (Figure 6C). Notably, those bacteria that were found to be differential by several methods between C, AD and CRC groups correlated with the same metabolite classes that were found to be mostly differential and discriminant between sample groups. Thus, *Fusobacterium* was found to present the most correlations of any bacteria, specifically with cholesteryl esters and sphingomyelins metabolite classes. This same genus clustered with other reported altered genera (*Gemella, Parvimonas Peptrostreptococcus* and Erysipelotrichaceae genera) that were also found to be positively correlated with the same metabolite classes. Regarding these metabolites, they were found also to correlate negatively with genera found to be decreased in CRC patients (*Coprococcus, Dorea, Blautia*). Apart from the mentioned metabolite classes, genera decreased in the CRC group was also negatively associated with diacylphosphatidylcholines (DAPC). Interestingly, we also observed a trend with triacylglycerol species and *Desulfovibrio* and *Synergistes* genera, which were also negatively correlated, suggesting a regulation of triacylglycerol metabolislm by these bacteria. Finally, an opposite trend was also identified for *Pyramidobacter* and *Roseburia*, in a way that for those metabolites that positively correlated with *Roseburia* did also correlate negatively with *Pyramidobacter*, mostly being triacylglycerols and DAPCs. Complete correlations plot can be found as Appendix A.

We have also applied Procrustes analysis [26] to identify the global similarities between both datasets, both considering all the samples and analyzing them by clinical category. In order to facilitate the comprehension of the results, the comparison was performed using each dataset corresponding eigenvalues. When comparing all samples, microbiome and metabolomics showed to be quite similar (*p*-value 0.001), although the correlation score (RV) was low (RV 0.30) (Appendix A). Separately, C, AD and CRC showed higher similarities between the microbiome and metabolomics data than when analyzed together (Appendix A), with RV scores ranging from 0.4 to 0.5 and all *p*-values being significant. Notably, microbiome PCoA for both AD and CRC sample groups (Appendix A) showed less diversity between individuals than the C group.

#### 2.4.2. Microbiome: Metabolomics Predictive Model

Finally, we decided to combine both microbiome and metabolomics data to generate a LASSO logistic model to test the potential predictive capability of this combined molecular fingerprint. In order to keep the model as simple as possible and seeing that the metabolomics analysis revealed similar results as the one our group published previously [7], we used the 16 differential genera identified and the 6 metabolites used in the model we previously published.

While the microbiome fingerprint model alone worked, as well as, the combination of both datasets for both C vs. CRC and AD vs. CRC discrimination models, the predictive ability was lost when comparing C vs. AD sample groups. As described before, when FOB measurements were incorporated at the model the performance was slightly reduced (Figure 6D). C vs. CRC model microbiome model median AUC was found to be 0.887, slightly improving to 0.928 when combined with the metabolites fingerprint. AD vs. CRC model median AUC was 0.870, which improved to 0.923 when metabolites were added. Finally, C vs. AD model performance was negligible, with an AUC of 0.278, slightly improved to 0.297 when metabolites were added.

Seeing that the inclusion of FOB measurements in the models did not improve their performance, we decided to analyze the distribution of FOB among the distinct group samples. We observed that CRC was highly different from both C and AD, but the distribution of FOB measurements among AD samples was more widely so that it did not present significant differences with the C group (adj. *p*-value 0.068) (Figure 6E). This FOB levels distribution among AD patients may explain the reduction of the predictive capability of the models when FOB was included as a covariate.

## 3. Discussion

While FOB tests have been demonstrated to be efficient in randomized CRC screening trials, a requirement for better non-invasive biomarkers for CRC still exists, especially for the early stages of the disease, both the adenoma step and the initial CRC stages, where FOB measurements are not efficient. Thus, the aim of this study was to identify potential biomarkers, able to discriminate both AD and CRC from feces samples by combining metabolomics and microbiome data. Feces are proximal to the colorectal mucosa, thus we considered they could represent adequately the structural and metabolic alterations related to the disease progression. Because of the role of gut microbiota in shaping the final fecal metabolome, we considered the combined study of both microbiome and metabolomics data could be relevant for explaining the potential biomarkers identified. Also importantly, CRC development and progression has been recurrently associated with microbiota alterations [15,16,27,28,29,30,31,32,33,34,35,36,37,38,39,40,41,42]. This association was reaffirmed with the common altered bacteria identified between high FOB and low FOB measurements and C and AD vs. CRC groups.

CRC-related microbiome alterations have been suggested to be potentially useful as diagnostic biomarkers source. In fact, several studies have already focused on this aspect. Zeller et al. [43] provided evidence of this biomarker potentiality by generating highly-predictive and accurate models using up to 22 microbial taxa for CRC, validating their findings with different cohorts. Shah et al. validated these findings in a meta-analysis combining nine published 16S rRNA sequencing datasets, identifying the same microbial taxa despite technical differences between the studies [37]. All studies published, though, reported alterations related to CRC, but not for its previous stage, adenoma. More recently, some authors have studied the role of microbiota in the adenoma stage. While not significant differences respect to healthy controls were observed when analyzing stool samples [33], the analysis of tissue samples revealed distinct microbiota populations for control, adenoma and CRC samples [38]. 

It seems plausible to assume that microbiota plays a central role in the development of adenoma-like lesions and progression to CRC. To identify which bacterial functionalities may be leading this progression metabolomics—microbiota combination studies are useful. To our knowledge, three studies have been published that used metabolomics—microbiome data to infer potential metabolic alterations in CRC [30,31,32]. Two of them were performed with the USA population, while the third one with Japan individuals. Notably, for the three cases, the sample groups included both healthy and CRC individuals, but no adenoma sample group was included. These studies reported associations between specific bacteria and amino acids, and in the case of Yachida et al. [32] that also analyzed bacterial metabolites, alterations in methane metabolism were reported.

In our study, we have observed that the CRC microbiome population was richer than the other sample groups. As other authors suggest, this may reflect an overgrowth of harmful bacteria instead of a healthy gut microbiota [33], such as seen in our diversity analyses. This would explain also why when applying a more equilibrated diversity index, like Shannon’s one, the differences between CRC and other groups were not detected. Thus, the increased amount of different OTUs found in CRC patients (Figure 3C) seem to reflect only the apparition of pathogenic and CRC-related bacteria in CRC disease (Figure 3D), but not a general decrease on the other bacteria found in C and AD sample groups. This hypothesis is also supported by shotgun-sequencing in feces samples [33] and the Procrustes results, that showed less diversity in AD and CRC patient groups, that could be associated with the appearance of disease-associated bacteria in all patients. The fact that we could not find nearly any difference between C and AD microbiome populations in any of the analyses performed could be related to the samples used, although other authors have described this situation before [32,43]. Interestingly, the literature reported driver bacteria seem to be mucosa-adherent, while this requirement is not found for passenger bacteria. These mucosal-adherent bacteria could be offering the necessary interactions with stem cells promoting focal lesions, being thus a link with the environment [15]. Since driver bacteria should appear in AD disease stage, the lack of differences in stool samples between C and AD patients may be explained because of that, as other authors suggested [15]. Notably, alterations on the microbiome composition have been described for tissue biopsies in AD patients [36,38].

As has been recurrently described before, the most significant alteration we found for CRC microbiome was the relevant increase of *Fusobacterium*, an invasive and proinflammatory bacteria [34,44,45,46]. We also confirmed the positive association between disease stage and its previously described relative abundance [47,48,49]. While other authors have identified *Fusobacterium* to be elevated also in AD patients, we could not replicate these findings, as no statistical significance was reached, although a certain trend was observed (Figure 4A). This could indicate that AD patients present a high interindividual diversity in microbiome composition terms. *Fusobacterium* infiltrates the cell, which could also explain the lack of differentiation between C and AD sample groups, as we studied microbiome composition in stool samples and not in tissue, as other authors have done [29,38]. The role of *Fusobacterium* in tumoral development and progression has been demonstrated experimentally [50] to be mediated by its FadA adhesin (FadAc). FadAc adheres to E-cadherin, triggering this way both the invasion of the host’s cells by *Fusobacterium* and the activation of the β-catenin/Wnt signaling pathway, leading the first one to the cell proliferation and the second resulting on tumoral growth [51].

An inflammatory role for the enrichment of *Erysipelotrichaceae* in CRC patients has been observed previously, linking these bacteria to increased levels of TNF levels [52], being highly immunogenic. It is logical then to suggest a link between the increased abundance of *Erysipelotrichaceae* and the inflammation occurring in the tumor microenvironment. Moreover, its abundance has also been associated with known CRC-risk factors, such as high-fat, Western-like diets [53]. This would also explain the positive correlations found for two *Erysipelotrichaceae* genera bacteria with metabolite species from ChoE and sphingolipids families, supporting the hypothesis of a notable effect of microbiota in the fecal metabolome.

Interestingly, in our study we found that the bacteria *Adlercreutzia* has different abundance between AD and CRC patients what could be relevant as early CRC biomarkers. Although alterations on *Coriobacteriaceae* bacterial family have been reported for metabolic disorders related to cholesterol alterations [54], no alterations have been reported in early stages of the disease for *Adlercreutzia* genus. Instead, other members of this bacterial family have been reported increased in CRC patients, such as *Collinsella, Eggerthella, Olsenella* and *Slackia* [55,56]. *Adlercreutzia* is a bacterium known to produce equol from isoflavonoids consumed in the diet [57], and considered to be the most contributing bacterium to equol levels in host. Equol presence in the host is also associated with lower dyslipidemia levels and higher levels of high-density lipoprotein-cholesterol [58], thus suggesting a role for these bacteria in the host’s health. Equol has also been inversely associated with CRC risk in prospective studies [59], so that the increased abundance of *Adlercreutzia* in AD samples and its reduction in CRC patients could also be associated to this fact. Because of equol is produced from consumed isoflavonoids, these alterations on *Adlercreutzia* could be associated with different dietary habits of AD individuals, a factor for which we could not control in our study because of the absence of these data.

The elevated influence of the microbiome upon fecal metabolome was also stated by the number of integration tests we performed, finding a significant amount of individual correlations between bacterial genera and metabolites (Figure 6C and Appendix A) and multivariate similarities of both datasets with Procrustes analysis (Appendix A). Also, the inferred functionality of the 16S sequencing data suggests a potential bacterial origin for our identified metabolites, especially for those being differentially abundant in CRC patients. This microbiota role on the fecal metabolome has been studied before [30,31,32,60]. Among the associations found between bacteria and metabolites, those relating CRC increased genera and cholesterol metabolites and sphingolipids were of special relevance, as they were the most strong and significant ones. Fat-rich diets, with relevant amounts of cholesterol species, are associated with CRC development. Cholesterol can be processed to bile acids and steroid hormones, among others [61,62]. The CRC-specific microbiome can increase the synthesis of secondary bile acids, which are known to be carcinogenic, thus promoting CRC development and progression [63,64]. In this context, the relevant number of associations between cholesterol species and sphingolipids with genera increased in CRC patients supports this role of microbiome upon disease progression. 

Notably, the inferred functional capabilities of the different microbiome populations showed alterations in pathways related to carbohydrates degradation (glycolysis, TCA cycle) and fermentation, leading to a reduction of available SCFA, as described previously [60]. Importantly, SCFAs have an immunomodulatory role [65,66] that, due to their reduction by CRC specifics microbiome composition, is lost, contributing in this way to perpetuate an inflammatory tumor environment. Relevantly also, inferred metabolic functionalities identified methane related pathways to be increased in CRC microbiota, suggesting thus a more anaerobic bacterial population. This is supported also by the increase of anaerobic bacteria, such as *Peptostreptococcus, Peptococcus* and *Parvimonas* [67], in CRC patients. In fact, these increase in bacteria also could be supportive of the oral microbiome hypothesis, as these bacteria are commonly found in the skin and mucosal surfaces of the mouth and upper respiratory tract, apart from the gastrointestinal one [67]. This shift towards an increased abundance of methanogenic bacteria and archaea populations is also in discussion nowadays [68,69,70] and has been reported by other authors [32], although our methodological approach could not identify any archaeal species. It’s known that detection of archaea species requires of adapted DNA extraction protocols, specific primer sequences and it suffers from a lack of annotated archaeal genomes in most used databases for 16S bacterial centered sequencing studies, as reviewed in [71]. This could explain the lack of Archaea sequences in our study, as we focused on the bacterial population of the human microbiome communities, as presented in the Methods section. Relevantly, methanogen density is negatively correlated to butyrate concentration in feces [72], thus supporting also the microbiota-related inflammatory events that occur in CRC pathogenesis, due to the fact that methane-producing bacteria consume SCFA. Therefore, the microbiome alterations in CRC seem to suggest an increase in methanogenic organisms which, in turn, reduces the butyrate available levels, losing in this way the immunomodulation host’s capabilities, perpetuating inflammation in the tumoral microenvironment. In fact, butyrate levels are associated with CRC by a set of pathways, including the regulation of genes widely associated with cancers, such as VEGF, p53 or WNT [73]. Mechanisms by which bacteria may promote mutations on the oncogenic genes include bacteria-induced DNA alterations, metabolic and hormone alterations, chronic inflammation and reduction on bacterial products with anticancer effects, as reviewed in [15]. For *Fusobacterium*, for example, it has been proposed that promotes carcinogenesis by invading the host cells [50,74]. Relevantly, all the alterations we have identified in this study may fit in these pro-inflammatory and pro-carcinogenic pathways.

## 4. Materials and Methods

### 4.1. Clinical Samples and Study Population

The patients were recruited from the COLONPREDICT study (batch 1 and 2 samples) [22] and from the Biobank of the Instituto de Investigación Sanitaria Galicia Sur (samples‘ batch 3). In both cases, the cohort consisted of patients with gastrointestinal symptomatology referred for colonoscopy. Exclusion criteria for the patients’ cohort were: age under 18, pregnancy, patients with previous history of colonic disease, patients requiring hospital admission, patients whose symptoms had ceased within 3 months of evaluation, and patients who declined to participate after reading the informed consent form. Fecal samples were self-collected a week before from the colonoscopy by the patients from one bowl movement without dietary or drug restriction and delivered to the hospital. From the hospital, samples were delivered to the laboratory, aliquoted and stored at −80 °C in less than 4 hours.

### 4.2. UHPLC-MS Metabolomics Analysis

Metabolomics analysis was performed in collaboration with OWL metabolomics, as described elsewhere [7]. A UHPLC−time-of-flight (TOF)-MS-based platform was used to analyze chloroform/methanol extracts, including glycerolipids, cholesteryl esters, sphingolipids, primary fatty amides and glycerophospholipids among the identified ion features. The metabolite extraction procedure was as follows. Stools were lyophilized for 3 days by using a LyoQuest −85 instrument (Telstar, Woerden, Netherlands). Afterward, 15 milligrams of lyophilized stool samples were mixed with 45 µL sodium chloride (50 mM) and 450 µL chloroform/methanol (30:1) in 1.5 mL microtubes at room temperature. The extraction solvent was spiked with compounds not detected in unspiked human stool samples [SM(d18:1/16:0), PE(17:0/17:0), PC(19:0/19:0), TAG(13:0/13:0/13:0), Cer(d18:1/17:0) and ChoE(12:0)]. After brief vortex mixing, the samples were incubated for 1 hour at −20 °C. After centrifugation at 16,000× *g* for 15 min, 35 µL of the lower organic phase were collected and dried under vacuum, discarding the solvent. The dried extracts were then reconstituted in 1000 µL acetonitrile/isopropanol (1:1), centrifuged (18,000× *g* for 5 min), and transferred to vials for UHPLC-MS analysis on an Acquity-Xevo G2 QTof system (Waters Corp., Milford, MA, USA). Samples were randomly divided into three batches, which contained a maximum of 78 samples. Chromatographic method and mass spectrometric detection conditions were described by Mayo et al. [75]. Data pre-processing was processed using the TargetLynx application manager for MassLynx 4.1 (Waters Corp., Milford, MA, USA). The percentage of missing values was computed for each metabolite among all the samples, divided by class. Metabolites presenting more than 30% of missing values were removed from downstream analysis. Remaining missing values were imputed by taking the minimum value of the corresponding metabolites divided by ten. Data was finally log-transformed.

### 4.3. Metabolomics Data Analysis

Log-transformed data was used to compute several basic statistics measurements (mean, windsored mean, median, standard error of the mean, standard deviation, coefficient of variation, interquartile range, kurtosis and skewness indexes and Shapiro-test for normality assessment). For each pairwise comparison, the following difference tests were computed: *F*-test, Student’s *t*-test (including power calculation), Wilcoxon and fold-change (including robust fold-change). These measurements were later used to generate volcano-plots for each comparison and to perform the corresponding functional analysis, pathway mapping and functional enrichment tests. Multivariate analysis, including both PCA and PLS-DA analyses, was performed with SIMCA-P+ 12.0.1 (Umetrics AB, Umeå, Sweden).

Direct metabolites were mapped to pathways with IMPaLA webtool [76]. Genes related to each differential metabolite were obtained from KEGG [77] and HMDB [78] databases using *ad-hoc* R and Python scripts. Enrichment for bacterial pathways was performed with *FELLA* [79] R package, using *E. coli* as a reference database. All stats computations were using stats, psych [80] and OptimalCutpoints [81] R packages. Correlation analysis was performed with corrplot [82], R package.

### 4.4. Fecal DNA Extraction

Half of the lyophilized feces aliquots were used for DNA extraction. The initial amount of lyophilized feces was weighted in each case. Fecal DNA was extracted using the commercial kit PSP^®^ Spin Stool DNA Kit (STRATEC Molecular, Birkenfeld, Germany), following the manufacturer’s recommendations. Briefly, lysis buffer was added to lyophilized feces samples and mechanical lysis step was done with zirconium beads, using Precellys^®^ equipment (Bertin Instruments, Montigny-le-Bretonneux, France), 1 cycle of 50 s. The supernatant was recovered and centrifuged with InviAbsorb resin to remove impurities. Then, the supernatant was recovered again and passed through a column filter with binding buffer and washed twice with ethanol buffer. Finally, DNA was eluted in 100 µL of elution buffer and stored at −20 ℃ until further processing. The DNA concentration and quality were assessed with Nanodrop^®^ equipment (Thermofisher, Waltham, MA, USA) reporting DNA concentration (ng/µL), A260, A280 absorbance and A260/280 and A260/230 ratios.

### 4.5. 16S rDNA Amplification and Sequencing

Variable regions V1 and V2 of the 16S rRNA gene were amplified using the primer pair 27F-338R in a dual-barcoding approach [83]. DNA was diluted 1:10 prior PCR and 3 µL of this dilution were finally used for amplification. PCR-products were verified using the electrophoresis in agarose gel. PCR products were normalized using the SequalPrep Normalization Plate Kit (Thermo Fischer Scientific, Waltham, MA, USA), pooled equimolarily and sequenced on the Illumina MiSeq v3 2 × 300 bp (Illumina Inc., San Diego, CA, USA). Demultiplexing after sequencing was based on 0 mismatches in the barcode sequences. Forward and reverse reads were merged using the FLASH software, allowing an overlap of the reads between 250 and 300 bp [84]. To eliminate low-quality sequences, the data were filtered by removing sequences with a sequence quality of less than 30 in less than 95% of the nucleotides. Chimeras were removed with UCHIME [85].

QIIME2 [86,87] was used to perform the diversity indexes measurements and the taxonomical annotation. In the QC steps of QIIME2 DADA2 [88] was used; for taxonomical annotation GreenGenes v(13_8) database was used as reference, at 99% OTUs similarity threshold.

Diversity and taxonomical measurements and OTU table obtained from QIIME2 pipeline were uploaded to R (https://cran.r-project.org) and analyzed with vegan [89], phyloseq [90] and ggplot2 packages.

SIAMCAT [91] and ALDEx2 [24] tools were used to study the differences at genus level between the three sample groups, in a complementary fashion in order to identify potential biomarkers for each disease stage. We also used SIAMCAT to test the potential influence of potential confounding factors such as the sample batch, gender and/or FOB measurement. The list of differential bacteria was later used to generate a LASSO model in order to evaluate the predictive capability of these bacterial genera with *ROCR* [92] package. PICRUSt2 [93] was used to infer the potential metabolic functionalities of the distinct microbiome populations.

### 4.6. Metabolomics—Microbiome Data Integration

#### 4.6.1. HAllA

“Hierarchical All-against-All significance testing” [25] was used to identify potential correlations between individual metabolites and specific bacteria genera. Relative abundance genera dataset and log-normalized identified metabolites peak-intensities were used as input for the HAllA pipeline, specifying Spearman’s correlation index for the measurements.

#### 4.6.2. Procrustes

Procrustes analysis was applied upon CLR-normalized microbiome data and metabolomics normalized data. In order to compare different data types, the Procrustes analysis was performed upon the Euclidean distances of eigenvalues of each dataset using vegan R package [89], as described elsewhere [94].

#### 4.6.3. MixOmics

DIABLO [95] pipeline was used for the integration of both microbiome and metabolomics datasets. CLR-normalized microbiome data and log-normalized metabolomics dataset were used for the integration of datasets.

## 5. Conclusions

To summarize, the current study integrates metabolomics and microbiome data, which allowed the generation of high-performance combined predictive models and provided with a biological framework for the metabolite changes observed. Several data also confirm the microbial role on the fecal metabolome composition, thus supporting the hypotheses for which CRC is associated with microbiome composition alterations in humans.

## Figures and Tables

**Figure 1 cancers-12-01142-f001:**
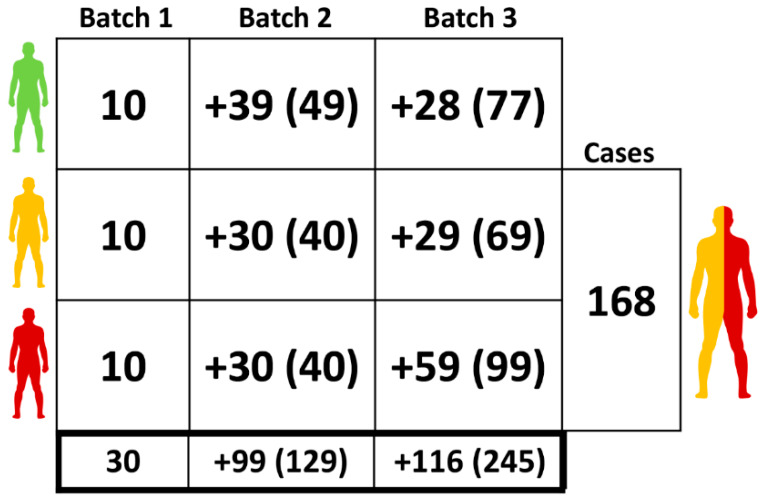
Distribution of samples between the different sample batches. Green human silhouette indicates the CONTROL group (C), yellow one ADENOMA one (AD) and red COLORECTAL CANCER group (CRC). The three batch samples are indicated at the top of each column and the total number of samples per batch in the final row. The *n* per group per batch is represented in cells, while between parentheses the cumulative number of samples per group is shown.

**Figure 2 cancers-12-01142-f002:**
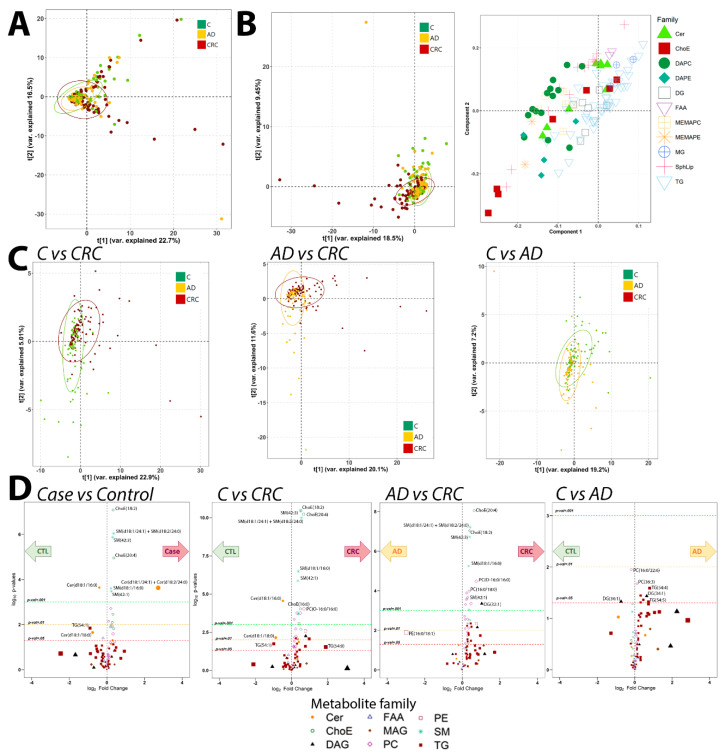
Multivariate and univariate analyses of the metabolomics data. Points are colored depending on their clinical classification. PCA scores plot (**A**), PLS-DA scores and loadings plot, with point shape and color depending on the metabolite class (**B**) and pairwise PLS-DA scores plots, from left to right: C vs. CRC, AD vs. CRC and C vs. AD comparisons (**C**). Volcano plots for the distinct comparisons of sample groups, with points shape and color depending on the metabolite class (**D**).

**Figure 3 cancers-12-01142-f003:**
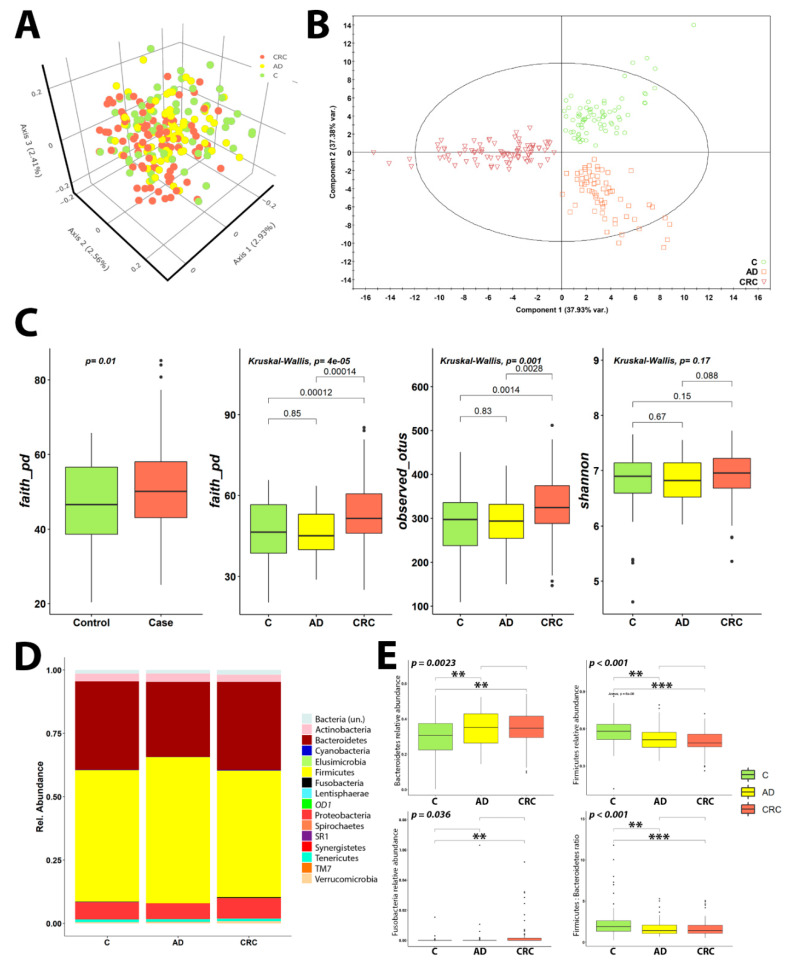
Diversity and taxonomical measurements of the microbiome data. Scores plot for the PCoA analysis performed upon Bray-Curtis distances of the microbiome. Points are colored depending on the sample clinical status (**A**). PLS-DA analysis of the Bray-Curtis distances. Point shapes and colors depend on the clinical status of samples (**B**). Alpha-diversity measurements of the distinct sample groups, Faith’s Phylogenetic Distance for both Case-Control and C-AD-CRC comparisons, Observed OTUs and Shannon diversity index for C-AD-CRC comparison (**C**). Mean relative abundances for all identified phyla per sample group stacked barplot, with different phyla colored differentially (**D**). Distribution of phyla relative abundances on the most differential ones, Bacteroidetes, Firmicutes and Fusobacteria per sample group. Differences in the ratio of Firmicutes to Bacteroidetes relative abundances per sample groups. *p*-Value scores are indicated in the following manner: < 0.05, * < 0.01, *** < 0.001 (**E**).

**Figure 4 cancers-12-01142-f004:**
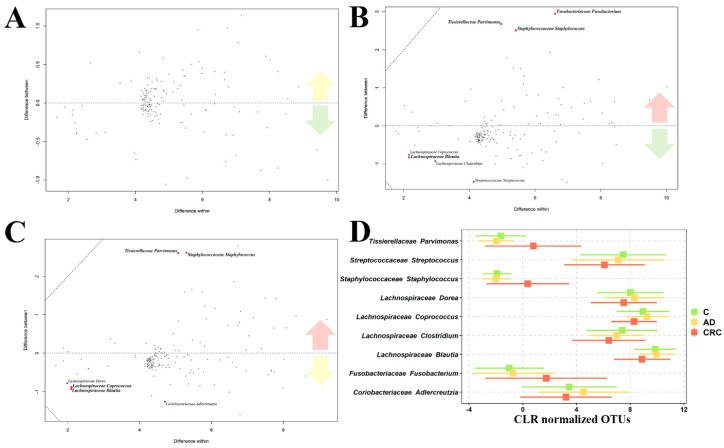
ALDEx2 results for the three comparisons: C vs. AD (**A**), C vs. CRC (**B**) and AD vs. CRC (**C**). Difference between sample groups abundances is depicted in the vertical axis, while horizontal one depicts the differences within each sample group. Grey points represent abundant non-differential features, black points the non-differential rarely abundant features, blue dots the features identified as significantly different by one test (*t*-test or Wilcoxon) and red ones the significantly different features identified by both tests. Distribution of CLR-normalized abundances of the significantly differently abundant bacteria identified by ALDEx2 methodology in the three sample groups (**D**). The central square indicates the mean of the distribution, while bars indicate the standard deviation of the distribution.

**Figure 5 cancers-12-01142-f005:**
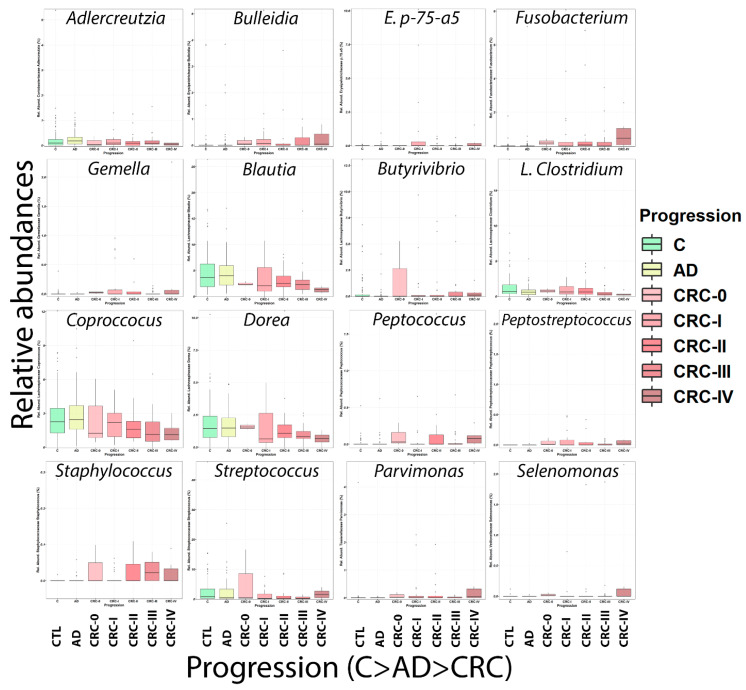
Relative abundance per sample group and CRC stage group of the 16 differentially abundant genera (n_C_ = 74, n_AD_ = 62, n_CRC-0_ = 3, n_CRC-I_ = 22, n_CRC-II_ = 22, n_CRC-III_ = 30 and n_CRC-IV_ = 6).

**Figure 6 cancers-12-01142-f006:**
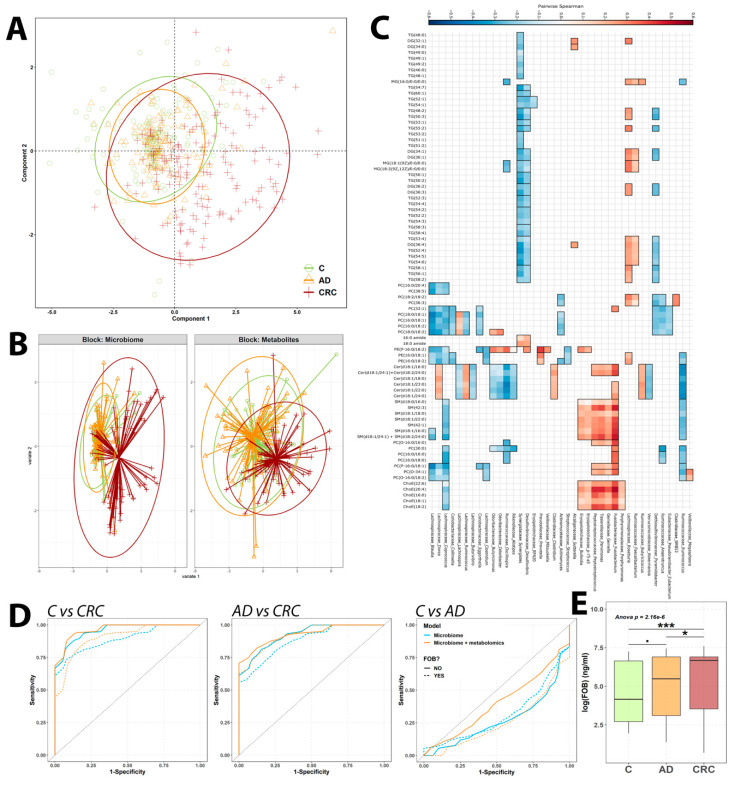
Multi-omics results summary (**A**–**C**) and metabolomics-microbiome fingerprint LASSO models ROC curves (**D**). mixOmics analysis for the combination of both metabolomics and microbiome data. Block sPLS-DA analysis scores plot for the 3 sample group. Point color and shape depend on the sample clinical status. Ellipses represent the 95% confidence per sample group (**A**); Contribution of each omics dataset to the distribution of the samples depicted in the block sPLS-DA scores plot. Point shape and color depend on the sample clinical status. Lines represent the distance from each sample to the centroid of the corresponding sample group. Ellipses represent the 95% confidence region (**B**); 50 strongest associations resulting from HAllA results, depicting the correlations between bacterial genera, displayed in the vertical axis, and individual metabolites, represented in the horizontal axis. The red color indicated positive correlation values, while blue color negative ones. Only significant correlations have been painted. Correlations have been also clustered depending on the correlation trend so that the genera that correlate with the same metabolites are depicted together (**C**); Median ROC curves for the 10,000 population iterations of the microbiome and microbiome-metabolomics combined models. Line color indicates the data used (blue microbiome, orange combination) and line type the inclusion of FOB measurements on the model (solid without FOB, dashed with FOB) (**D**); FOB measurements distribution per sample group. FOB measurements are indicated in log-scale and ANOVA *p*-value of the differences between sample groups is indicated. Pairwise comparisons significance levels are indicated as follows: <0.1, * < 0.05, ** < 0.001 and *** < 0.001 (**E**).

**Table 1 cancers-12-01142-t001:** Differences between the identified phyla mean relative abundances per sample group comparison. ANOVA values are indicated in the first column, while Tukey’s HSD test *p*-values are indicated per each pairwise comparison in the other columns. In black, non-significant differences, in red are indicated the *p*-values between 0.05 and 0.01, in yellow the *p*-values between 0.01 and 0.001 and in green the *p*-values < 0.001.

BACTERIAL PHYLUM	PR(>F)	C-AD	CRC-AD	CRC-C
***K__BACTERIA.__***	0.450	0.974	0.630	0.461
***K__BACTERIA.P__ACTINOBACTERIA***	0.577	0.827	0.916	0.548
***K__BACTERIA.P__BACTEROIDETES***	**0.002**	**0.009**	0.998	**0.006**
***K__BACTERIA.P__CYANOBACTERIA***	0.247	0.695	0.722	0.216
***K__BACTERIA.P__ELUSIMICROBIA***	0.396	0.528	0.994	0.418
***K__BACTERIA.P__FIRMICUTES***	**<0.001**	**0.002**	0.449	**<0.001**
***K__BACTERIA.P__FUSOBACTERIA***	**0.036**	0.634	0.285	**0.030**
***K__BACTERIA.P__LENTISPHAERAE***	0.086	0.865	0.283	0.085
***K__BACTERIA.P__OD1***	0.096	1.000	0.168	0.145
***K__BACTERIA.P__PROTEOBACTERIA***	0.165	0.769	0.519	0.146
***K__BACTERIA.P__SR1***	0.450	1.000	0.544	0.517
***K__BACTERIA.P__SPIROCHAETES***	0.951	0.957	0.958	1.000
***K__BACTERIA.P__SYNERGISTETES***	0.567	0.564	0.691	0.967
***K__BACTERIA.P__TM7***	0.746	0.726	0.879	0.946
***K__BACTERIA.P__TENERICUTES***	0.920	0.990	0.967	0.915
***K__BACTERIA.P__VERRUCOMICROBIA***	0.549	0.965	0.559	0.705

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
