# Peer review of "Integrative Analysis of Fecal Metagenomics and Metabolomics in Colorectal Cancer"

_cancers, 2020, doi:10.3390/cancers12051142_

Round 1

Reviewer 1 Report

Title:  Integrative Analysis of Fecal Metagenomics and Metabolomics in Colorectal Cancer

Authors: Clos-Garcia et al.

In this work, Clos-Garcia and collaborators characterize the metabolome and microbiome of fecal samples a selected samples of AD/CRC patients along with healthy controls. Microbiome is analyzed using 16S v1-v2 approach, while untargeted UHPLC-MS is used for metabolomics. Notably, metabolomic and microbiome data is integrated using several state-of-the-art data analysis methods, and provide an additional functional track regarding CRC-microbiome interaction that warrants further research.

Reported results add up to knowledge base on the relationship between the gut microbiome and CRC. The manuscript reads well and methods are thoroughly detailed. Perhaps, the manuscript could benefit from a reorder in how results from different data analyses methods are reported.

I have several minor issues that I think need to be assessed before publication:

  • There is no sampling information in methods. I understand that these are feces, but the sampling method need to be reported.
  • Data PERMANOVA results should be extende to report the magnitude of the effect and not only significance.
  • Could authors plot bacterial loadings in Figure 3B (regarding PLS-DA on microbiome data)
  • I think that Figure3d should report sample-level stacked barplots. Summarized barplots give no information of variance.
  • Regarding Table 1 I am not sure on how permanova results were obtained for single phylum. This is no
  • Authors report that no archaea was present in this dataset. Discussion section mentions the reported associations of archaea, butyrate and inflammation. Therefore, undetection of archaea in fecal samples in any of the clinical groups is relevant, since is arguably one of the main groups possibly contributing to CRC development. This should be discussed further.
  • It is surprising that FOB information decreases the accuracy of the LASSO model. If the new variable does not improve the model, LASSO algorithm should just simply discard it.
  • No method info on PLS-DA is provided. Also, could ellipses be added to PLS-DA plots to project differentiation among groups?
  • Authors mention th C vs AD pairwise PLS-DA on metabolome did not differentiate. I’d would agree on that in the case of C+AD+CRC analysis but it looks like CvsAD also provide some difference. It’d would also be great to see some measure of these accuracy of these PLS-DA models.
  • While I applaud the combination of several data analysis methods, it is a bit confusing to follow as the results are often intermingled in the paper and in the figures. For instance in figure 6d, I thought that the ROC curves corresponded to the Diablo analysis, and did not understood how the FOB info was included.
  • Data transformation is also a bit confusing. CLR-normalized microbiome detailss used for mixOmics integrative block sPLSDA, but not for HAIIA for which RA were used. I am not certain about HAIIA method, but since the compositionality of microbiome data may hamper regression results, wouldn’t it be advisable to use CLR-transformed data for such HAIIA exploration? Another example would be figure 5 were RA abundance are plotted on selection made over CLR-transformed data.

Reviewer 2 Report

This is a well articulated study and offers interesting insights into the links between fecal microbiota and colorectal cancer. The methodology and results are well explained and I have just a few minor suggestions to add.

1. 2.2 Metabolic analysis- The authors have used PLS-DA analysis to discriminate different groups. I believe it would be more informative if the authors were to include a tSNE comparison of these datasets, to prove a better overview of the distribution and distinction of data points.

2.  Likewise this reviewer would like to see a tSNE map of the mixomics results in 2.4.1 and Figure 6.

3. The authors have discussed the poential mechanisms linking fecal microbiota and cancereous phenotypes via multiple pathways. However, the study would have a better impact if the authors could provide an illustrative example on a specific pro-inflammatory or pro-carcinogenic pathway that is strongly influenced by the fecal bacteria.

4. Minor language editing recommended- Introduction "However, understanding on 68 how the bacteria composition impact in host metabolism is still missing and further investigation 69 connecting microbiome and metabolome are needed." This sentence (and a few similar instances throughout the text) could be rephrased.
